# Autonomic Rebound Following Maximal Exercise in Bodybuilders and Recreationally Active Participants

**DOI:** 10.3390/sports12060143

**Published:** 2024-05-25

**Authors:** Brian Kliszczewicz, Gabe Wilner, Andre Canino, Pedro Chung, Abigail Nickel, Keilah Vaughan, Cherilyn McLester, Robert Buresh

**Affiliations:** Exercise Science and Sport Management, Kennesaw State University, Kennesaw, GA 30144, USA; gwilner@students.kennesaw.edu (G.W.); acanino@students.kennesaw.edu (A.C.); pchung6@students.kennesaw.edu (P.C.); amm1096@gmail.com (A.N.); kvaugh42@students.kennesaw.edu (K.V.); cmclest1@kennesaw.edu (C.M.); rburesh@kennesaw.edu (R.B.)

**Keywords:** bodybuilding, heart rate variability, resistance training

## Abstract

The off-season for natural bodybuilders (BB) is characterized by increased training loads and fluctuations in caloric intake, which may lead to insufficient recovery. The autonomic nervous system (ANS) plays a pivotal role in recovery. The purpose of this study was to evaluate resting ANS activity and recovery following a maximal exercise bout in off-season BB and compare them to those of recreationally active individuals. Fifteen males participated; 7 recreationally active (RA) (24.6 ± 2.1 years, 81.1 ± 10.8 kg) and 8 BB (21.8 ± 2.9 years, 89.3 ± 13.0 kg). Each performed a graded exercise test. Heart rate variability (HRV) was measured at rest and during a 45 min recovery period. HRV was analyzed as: root mean square of successive differences (lnRMSSD), standard deviation of normal-to-normal sinus beats (lnSDNN), high frequency (lnHF), low frequency (lnLF), and the ratio of low frequency to high frequency (lnLF/lnHF). A one-way ANOVA showed no differences for any resting marker of HRV, HR, and HR recovery. A significant depression in all markers of HRV was observed in the BB group at the 15 min point, and no recovery was observed before 45 min when compared to RA. The results of this study demonstrated depressed HRV recovery following the graded exercise test in BB when compared to the RA group.

## 1. Introduction

Bodybuilding is an established competitive sport that involves heavily structured weightlifting routines and diets that emphasize muscle building or fat-burning, depending on the competition phase [1]. Currently, there is little information related to bodybuilding and physiological stresses due to the purported use of performance-enhancing drugs (PEDs), presenting a litany of health, ethical, and study design concerns [2]. Even less is known about natural bodybuilding, which has recently seen a growth in popularity [3]. This population pursues similar training goals and competitive outcomes as those of traditional bodybuilding, but without the use of PEDs. A normal bodybuilding season is commonly broken into two distinct phases: 10–20 weeks in the ‘bulking phase’ (off-season) and 8–12 weeks in a ‘cutting phase’ (prep) [4,5,6]. Each phase has unique nutritional requirements, volumes of aerobic exercise, and approaches to resistance training methodology. The contest prep phase has been the most evaluated [3,4,5,6], with an emphasis on training approaches; however, the larger training loads and fluctuations in caloric intake experienced during the off-season phase may be of equal or greater importance, given that there is a higher potential for insufficient recovery between workouts and overtraining [7]. Overtraining occurs when the physiological demands of an exercise program outweigh the ability of the body to adapt, negatively impacting several physiological systems, including the neuroendocrine, immunological, cardiovascular, and musculoskeletal systems, respectively [8]. The activity of the autonomic nervous system (ANS) plays a pivotal role in the acute and prolonged recovery from exercise challenges [9,10] and can become impaired by overtraining [11]. In some instances, prolonged disruptions in ANS balance, as a result of intense exercise, have been linked to the increased likelihood of sudden cardiac events [12,13].

A non-invasive and relatively cost-effective evaluation of ANS activity can be derived through the measurement of heart rate variability (HRV). HRV can be measured and quantified with various devices by measuring the timing between consecutive R-R intervals [14]. Though the ANS comprises sympathetic and parasympathetic (PNS) branches, only the PNS has been reliably reflected through HRV [15]. There are several markers of HRV, but only a few are commonly accepted as markers of PNS activity and are sensitive to physical activity or stress: the root mean square of successive differences (RMSSD) and the frequency domain index of high frequency (HF) [14,15]. It has been well-established that bouts of physical activity facilitate the depression of HRV, while the cessation of activity results in its rebound [15,16,17]. The magnitude and duration of this depression and rebound are influenced by an array of factors, including cardiovascular fitness and physiological recovery/readiness [15]. HRV rebound can be a useful tool to gauge transient autonomic nervous system stress following a bout of exercise. 

Current research on bodybuilders focuses on body composition outcomes and training variables such as nutrition and supplementation [4,5,6]. The physiological changes that occur within the off-season phase and their effects on the ANS and subsequent rebound are not well understood. A better understanding of the physiological outcomes during this phase may provide additional insight into potential untoward effects of high-volume resistance training. Additionally, research in this area may provide important information regarding safe and effective offseason training approaches for BB. Therefore, the purpose of this study was to evaluate natural bodybuilders who are within the off-season phase to assess resting autonomic activity and its recovery following a maximal bout of exercise through HRV and compare them to those of recreationally active individuals.

## 2. Materials and Methods

### 2.1. Experimental Design

Data collection occurred in the Kennesaw State University Exercise Physiology Lab during a single visit between the hours of 6 a.m. and 11 a.m. Upon arrival, participants were given the opportunity to review the informed consent form and ask any questions prior to withdrawing or signing. Prior to the collection of data, informed consent was obtained from all subjects involved in the study. After consent, a health history questionnaire and participant demographics were collected. Body composition was then collected via stadiometer and dual energy x-ray absorptiometry (DEXA). A capillary finger stick was then used to assess fasting metabolic profile (glucose and lipid profile). Following this, participant pre-exercise HRV was measured through electrocardiogram (ECG) (Finapres NOVA). A modified ramp treadmill graded exercise test (GXT) was performed; then, participants had recovery HRV measured for 45 min. A total of fifteen males participated in this study: eight natural bodybuilders (BB) and seven recreationally active (RA) individuals. Participant characteristics can be found in Table 1. The one-way ANOVA showed no significant differences in height or weight between the BB and RA groups; however, significant differences in BF% were observed for DXA (*p* = 0.004), with BB exhibiting a lower BF% when compared to RA, while age was shown to be trending (*p* = 0.055).

### 2.2. Participants

The study was conducted in accordance with the Declaration of Helsinki, and approved by the Institutional Review Board of Kennesaw State University IRB-FY23-219, 19 January 2023 for studies involving humans. A total of fifteen apparently healthy males (23.0 ± 2.9 years, 177.6 ± 8.5 cm, 85.4 ± 12.4 kg) participated in this study, including eight natural male bodybuilders (BB) (21.8 ± 2.9 years, 176.6 ± 7.2 cm, 89.3 ± 13.0 kg) and seven recreationally active (RA) males (24.6 ± 2.1 years, 178.6 ± 9.6 cm, 81.1 ± 10.8 kg). The inclusion criteria for the BB participants required them to have no PED usage, which was verbally confirmed upon recruitment as well as during data collection, actively train in their off-season, have the intention of putting on muscle mass, and have completed at least one full bodybuilding training cycle including an “off-season” (bulking phase), a “preparation” (cutting phase), and a “maintenance” phase. The inclusion criteria for the RA participants required consistently performing 150 min of moderate aerobic exercise each week and resistance training at least two days each week over the past 6 months.

To ensure that participants could perform vigorous activity, as defined by the guidelines of the American College of Sports Medicine, a health history questionnaire was administered and reviewed by primary investigators. Any individual who reported having orthopedic conditions, cardiovascular, pulmonary, or metabolic disease was excluded from the study. Additionally, anyone with recent musculoskeletal injuries (e.g., muscle tears or fractures) over the past 6–12 months or current physical limitations was also excluded. Participants were recruited under their claim that they were not taking any PEDs or medications that influence metabolic control. Recruitment took place via word of mouth from the local metropolitan area and was targeted towards college students on or around campus who fit the inclusion criteria. Prior to all sessions, participants were asked to wear clothing suited for exercise that was light and comfortable, fast for a minimum of eight hours (10–12 encouraged), avoid exercise and alcohol for 24 h, and avoid caffeine consumption for 12 h. 

### 2.3. Pre-Heart Rate Variability, Anthropometric Measurements, and Graded Exercise Test (GXT)

Anthropometric measurements were collected via an electronic physician’s scale (Tanita WB 3000, Arlington Heights, IL, USA) to measure height (cm) and weight (kg). The analysis of body composition was assessed through dual-energy X-ray absorptiometry (DXA) (GE Lunar iDXA, GE HealthCare, Chicago, IL, USA). All scans were performed according to the manufacturer’s guidelines including the removal of shoes and any metal (e.g., jewelry). For the duration of the scan, subjects were supine and instructed to remain motionless with their arms by their side. 

Following body composition measures, a 10 min resting HRV was collected. For all time points of HRV measurement, participants were placed in a quiet, dimly lit room where they were asked to remain still while in a supine position. Recordings were collected through an electrocardiogram-capturing device (Finapres, Enschede, The Netherlands). Electrodes were placed in a modified 3 lead configuration with electrodes placed on the shoulders (crevice between anterior deltoid, pectoralis major, and clavicle) and hips (one inch medial to the anterior superior iliac spine). Prior to placing the electrodes, an abrasive gel was applied to clean off any loose skin or oil, and then the skin was cleaned with acetone. Once the electrodes were placed, participants were asked to lie in a supine position for 10 min while their electrocardiogram was recorded. Heart rate variability was analyzed in five-minute segments, with the first 5 min being discarded. The last 5 min of the baseline recording was used (PRE). 

Aerobic capacity (VO_2_ peak) was assessed during the lab visit through a graded exercise test (GXT) on a treadmill (Woodway USA, Waukesha, WI, USA) using a modified ramp protocol at a self-selected speed that remained constant while the grade increased by one percent until volitional exhaustion was achieved. Expired gas fractions were sampled using a metabolic system (Parvo Medics Cart), per the manufacturer’s specifications. VO_2_ peak was considered to be sufficient if volitional exhaustion coincided with a respiratory exchange ratio of ≥1.10, and/or a heart rate (HR) within 10 bpm of age-predicted HRmax (208 − 0.7 × age). After completion of the GXT, participants were given a 1–2 min cooldown during which the treadmill was gradually slowed, the headgear and gas collection apparatus removed, and the participants were returned to the HRV testing table. 

### 2.4. Post GXT Heart Rate Variability 

Following the active cool-down period of the GXT, a 45 min HRV recovery period was recorded. The segments of the 45 min HRV period were also broken down into five-minute intervals, 5–10, 10–15, 15–20, 20–25, 25–30, 30–35, 35–40, and 40–45 min to be analyzed. Kubios software (Kubios V 2.2, Joensuu, Finland) was used to analyze the time domain and frequency domain measures. The artifact noise was filtered through a piecewise cubic spline interpolation method using a “low artifact correction” with a sensitivity set to identify any R-R abnormality ±0.35 s compared to the local average through a function available in the Kubios software [18,19]. In order to avoid distortion of analysis, any segments containing three or more irregular R-R intervals (e.g., artifact or ectopic beats) [20] were excluded from analysis. 

The time domain measures of HRV chosen for the current study include the root mean square of successive R-R differences (RMSSD), which is a widely accepted and documented marker of PSNS activity (22). Time domain analysis occurs through the process of converting R-R intervals into a graph called a tachogram where the Y-axis represents the time (ms) between consecutive intervals and the X-axis represents the number of beats over time. Additionally, frequency domain measures of low frequency (LF), high frequency (HF), and the low frequency to high frequency ratio (LF/HF) were analyzed using the Fast Fourier Transformation technique (FFT). The FFT analyzed the different frequencies and separated them into the categories of LF (0.04–0.15 Hz) and HF (0.15–0.4 Hz). Low frequency is established as a universal measure that indicates both PSNS and SNS activity, whereas HF and RMSSD reflect the PSNS activity [14,15].

### 2.5. Statistical Analysis

All data were entered and analyzed in SPSS 28.0.0.0 (190) software. A Shapiro–Wilk Normality Test was performed on all HRV data and found that normality was not violated, and data did not require log transformation. Measurements of lnRMSSD (ms), lnSDNN, lnLF (ms^2^), lnHF (ms^2^), and lnLF/lnHF were analyzed in Kubios (Kubios V 3.0, Joensuu, Finland). Pearson product correlations were used to assess potential relationships between resting heart rate measures, time domain measures, and frequency domain measures. Pearson product correlations were also used to assess any relationships between participant characteristics (i.e., height, weight, body comp, VO_2_ max, etc.). A one-way analysis of variance (ANOVA) was used to determine differences between the primary characteristics of BB and RA. The one-way ANOVA was also used to determine changes over the course of the HRV recovery period. The statistical significance was set to alpha < 0.05. Data are presented as the mean ± standard deviation (SD). 

## 3. Results

### 3.1. Cardiovascular

The one-way ANOVA revealed no significant differences between resting HR or HRmax between the two groups. Similarly, no differences were observed between VO_2_ peaks, regardless of the units the results were expressed in (mL/kg/min or mL/kgFFM/min). The results can be found in Table 2. 

### 3.2. Heart Rate Variability 

The one-way ANOVA showed no significant differences for resting HRV time-domain measures of lnRMSSD (*p* = 0.382) and lnSDNN (*p* = 0.342); and frequency domain measures lnHF (*p* = 0.354) and lnLF (*p* = 0.384). All time points following the GXT were significantly less than the PRE values for all HRV markers (*p* < 0.05). There were group-based differences in the recovery at various time points for each marker of HRV. The lnRMSSD was significantly lower in the BB group at 20–25 (*p* = 0.028), 25–30 (*p* = 0.021), and 40–45 (*p* = 0.018) (Figure 1A); the lnSDNN was significantly lower in the BB group at 20–25 (*p* = 0.025), 25–30 (*p* = 0.012), 30–35 (*p* = 0.027), 40–45 (*p* = 0.022) (Figure 1B); the lnHF was significantly lower in the BB group at 15–20 (*p* = 0.038), 20–25 (*p* = 0.009), 30–35 (*p* = 0.018), 40–45 (*p* = 0.001) (Figure 2A); the lnLF was significantly lower in the BB group at 15–20 (*p* = 0.031), 25–30 (*p* = 0.041), 30–35 (*p* = 0.043), 35–40 (*p* = 0.009), and 40–45 (*p* = 0.036) (Figure 2B). The lnLF/lnHF ratio was significantly different at 20–25 (*p* = 0.012) and 40–45 (*p* = 0.001) (Figure 3A), while no differences were observed with the HR Recovery (Figure 3B). 

## 4. Discussion

The primary findings of this study demonstrate a significant difference in the rate of recovery at various time points in all markers of HRV between the two groups, with the HRV in the BB group being more depressed. Additionally, with the exception of lnLF in RA, both the BB and RA groups failed to fully recover in all markers of HRV by minute 45, with the recovery for BB being significantly lower than that for RA. No differences in HR, or HR recovery were observed. This study provides insights into the cardiac autonomic differences between natural BB and RA, and expands upon the limited information related to the examination of vagal rebound following a maximal bout of exercise in a body-building population.

Perturbations in autonomic balance in the form of PSNS withdrawal and simultaneous increase in SNS activity over the course of an exercise bout are well established [21]. The responsiveness and degree of perturbation reflects the ANS ability to meet the demands of the exercise imposed on the body. However, when this shift in ANS balance is prolonged, there is an increased likelihood of untoward cardiac function (e.g., arrythmias) in those with underlying or undiagnosed conditions [22,23]. This study did not set out to determine these thresholds or occurrences, but rather to observe different patterns of ANS depression and rebound through HRV which provides an index of vagal activity. The magnitude and duration of the ANS disruption has been directly linked to exercise duration and intensity [16,17,21,24], with intensity being the greater influence [16]. Graded exercise tests, such as the one used in this study, have been shown to elicit relatively large changes in ANS activity [25], facilitating the evaluation of normal or abnormal responses. Both groups achieved similar VO_2_ peaks (ml/kgFMM/min) and peak HRs (Table 2) during the GXT. Following the bout, there was a nearly identical depression in HRV markers at the 5–10 min time point and the 10–15 min time point in the BB and RA groups. These findings suggest that participants experienced comparable exercise stimuli through the GXT, and other factors, such as training status/type or body composition of the participants may be more influential during the early stages (i.e., 15 min) of recovery.

The rate of vagal rebound following exercise is dependent on several factors such as the exercise modality and fitness status (i.e., VO_2_ max) [10]. During the first two timepoints following the GXT, no differences in the rates of HRV rebound were observed. Interestingly, by the 15–20 min time point, a significant difference in recovery status between the groups became apparent. This difference continued throughout the remaining timepoints, with recovery in BB being significantly blunted (Figure 1 and Figure 2). Generally, higher cardiorespiratory fitness (i.e., VO_2_ max) translates to a greater ability to recover PSNS activity [10]. Despite both groups achieving a comparable VO_2_ max, a gradual difference in rebound was seen. Though this study did not set out to determine mechanisms related to observed differences, a few may be hypothesized. The first possible explanation for the observed differences in rebound may be related to the BB group’s ‘off-season’ training. Although aerobic training data was not obtained from the BB group, it has been established that the off-season phase is typically characterized by significant reductions in aerobic exercise volume [4]. It has been postulated that training for events requiring higher levels of vagal control (e.g., long distance running) for prolonged periods of time may create a suitable stimulus for enhanced vagal recovery [26]. Therefore, the reduced aerobic training, or lack of prolonged vagal control common in the off-season phase, could have resulted in the observed delayed rebound. Performing some aerobic activity during this phase could be a useful technique to maintain aerobic conditioning and vagal control. Additionally, it has been shown that prescribing a light bout of aerobic exercise following periods of high-intensity training leads to accelerated ANS recovery [27]. Future studies should examine the use of aerobic prescription in off-season bodybuilders and its effect on vagal control. 

Several studies have reported changes in vagal activity as the result of overreaching and overtraining, with depressions in vagal activity at rest being among the most observed [7,28,29,30]. It was hypothesized that the BB group may be more susceptible to overtraining due to the increases in resistance training and volume that occurs during the off-season phase, which may be reflected through HRV markers. However, this was not the observed in the current study, in that none of the markers of resting HRV were significantly different between groups. What is less evaluated is the vagal rebound following a strenuous exercise bout to assess overreaching or overtraining. The current study did observe differences in vagal rebound, with BB being significantly delayed, which supports the initial hypothesis of potential overtraining. Importantly, determining if the observed difference in rebound was the result of fatigue related to overreaching or overtraining is not possible, but this information highlights a potential effect that should be further evaluated. As a point of consideration, a study by Hoshi et al. suggests that the clearance of lactate is a facilitating factor of sympathovagal reorganization during recovery following exercise [31], in that, above normal lactate levels blunt the reorganization and result in a delayed rebound. Though lactate was not measured in this study, the differences in training modality as well as the reduction in aerobic training for the BB group may be related to the observed differences in recovery. Future studies should examine rates of lactate clearance during the recovery period to determine if a relationship exists.

### 4.1. Practical Application

Overtraining and overreaching is a common issue amongst bodybuilders. Increasing training loads and frequency is a necessity for the desired outcomes for hypertrophy during the bulking phase [5], but increases the risks of burnout or injury [7]. Finding strategies to mitigate or even prevent overtraining can reduce lost training days or diminished workout capacity, which is commonly experienced [7]. The use of HRV to examine recovery status following bouts of exercise is a novel technique to indirectly monitor the return to homeostasis [10]. Monitoring HRV and deviations in rebound may provide needed insight for programing in order to prevent overtraining [28,29]. 

### 4.2. Limitations

This study involves a novel attempt to evaluate bodybuilders autonomic rebound and compare it to that of recreationally active participants. However, the study was not without limitations. First, bodybuilders should be evaluated during multiple training phases to assess the changes that may occur throughout a training cycle. Additionally, future studies should collect multiple baseline HRV recordings to establish a more robust baseline, rather than a single day. Finally, measuring lactate throughout the recovery period could provide additional context for the recovery differences observed. 

## 5. Conclusions

The results of this study demonstrated depressed HRV recovery following the graded exercise test in BB when compared to the RA group. Interestingly, the magnitude of HRV depression exhibited in these groups was identical and did not differ until the 15–20 min recovery mark, and remained throughout the 45 min recovery period. Both BB and RA groups achieved similar VO_2_ peaks, maximal heart rates, and heart rate recovery, despite the differing vagal recoveries. Overall, these findings suggest that the BB group during off-season training exhibited a depressed HRV recovery following maximal exercise when compared with recreationally trained males. Understanding and monitoring recovery status following training bouts may be a useful tool for the effective and safe prescription of exercise and should be further evaluated. 

## Figures and Tables

**Figure 1 sports-12-00143-f001:**
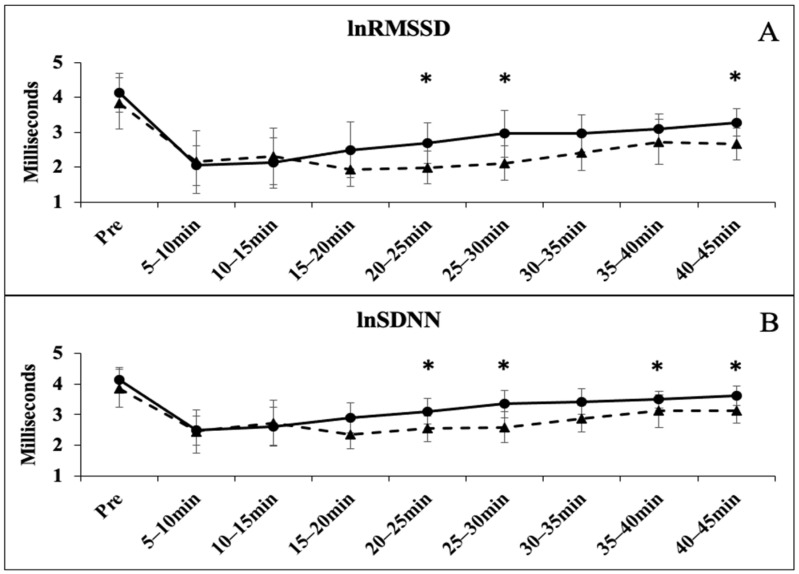
Time domain measures of heart rate variability before and after the graded exercise test between BB (triangle with dashed line) and RA (circle with solid line). (**A**) The time course of the natural log of RMSSD between BB and RA; (**B**) the time course of the natural log of SDNN between BB and RA. * denotes significant difference between groups.

**Figure 2 sports-12-00143-f002:**
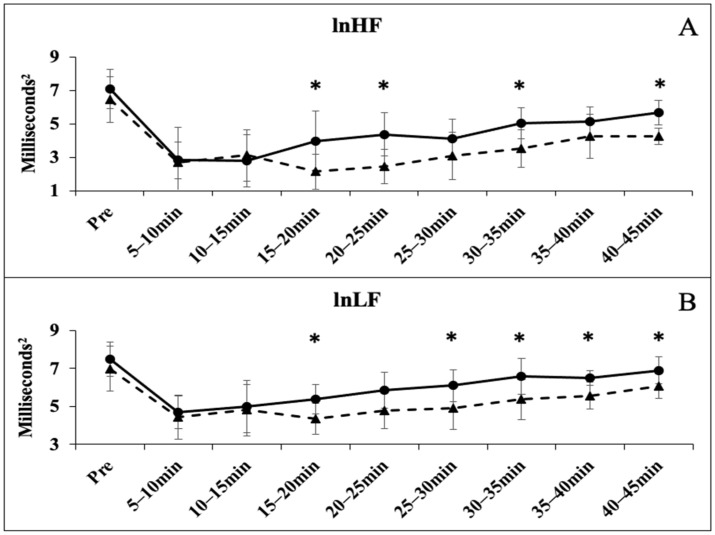
Frequency domain measures of heart rate variability before and after the graded exercise test between BB (triangle with dashed line) and RA (circle with solid line). (**A**) The time course of the natural log of HF between BB and RA; (**B**) the time course of the natural log of LF between BB and RA. * denotes significant difference between groups.

**Figure 3 sports-12-00143-f003:**
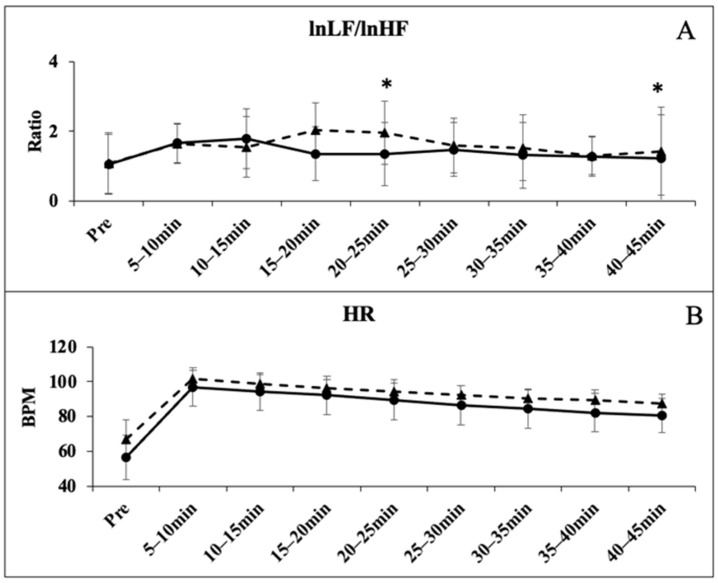
lnLF/lnHF ratio variability before and after the graded exercise test between BB (triangle with dashed line) and RA (circle with solid line) (**A**). Heart rate measurements before and after the graded exercise test between BB (triangle with dashed line) and RA (circle with solid line) (**B**). * denotes significant difference between groups.

**Table 1 sports-12-00143-t001:** Participant characteristics.

	BB Group Avg	SD	RA Group Avg	SD	*p*-Value
Age (years)	21.8	2.9	24	2.1	0.055
Height (cm)	176.7	7.2	178.6	10.4	0.685
Weight (kg)	89.3	3.0	81.1	10.8	0.211
DXA PBF (%)	13.4 *	4.9	22.0 *	4.5	0.004

***** = significant difference between groups.

**Table 2 sports-12-00143-t002:** Cardiovascular characteristics.

	BB Group Avg	SD	RA Group Avg	SD	*p*-Value
Resting HR (bpm)	65.3	11.3	56.4	12.6	0.123
Max HR (bpm)	185.9	8.9	187.6	7.7	0.337
VO_2_ Peak (mL/kg/min)	48.4	5.1	48.0	6.6	0.457
VO_2_ Peak (mL/kg FFM/min)	54.5	7.6	60.5	5.5	0.108

## Data Availability

Data will be made available upon acceptance.

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
