# Peer review of "Autonomic Rebound Following Maximal Exercise in Bodybuilders and Recreationally Active Participants"

_sports, 2024, doi:10.3390/sports12060143_

Round 1

Reviewer 1 Report

Comments and Suggestions for Authors

Dear authors,

this could potentially be an interesting work but could be improved in some details. Below are some comments that might be helpful

- In line 44: you mentioned for the first time the word “overtraining”. However, you consider this topic just in the discussion session “Several studies have reported changes in vagal activity as the result of overreaching  and overtraining” But what is overtraining from a muscle physiology point of view? It could be clearer to add a definition of “overtraining” in the introduction session to better understand what you refer to then in the discussion session.  a study from Bianco et al. (doi: 10.3390/jfmk4040068) could be useful for this purpose as it comprehensively describes an overview of the overtraining topic.

- Line 76: “Exercise Physiology Lab” where? Please, add this information

- Line 174: How did you determine the Sample size?  Please, add this information

- Line 194 and Table 1: there was a significative difference between the group’s ages.

- Line 273: Please, correct “th” in “the”

- Line 346-349: light bouts of aerobic exercise have been shown to accelerate ANS recovery “Who demonstrated this? Please add references and discuss this sentence.

 “and may be effective for supporting ANS recovery in bodybuilders during the offseason training. That is, it may be beneficial to perform some aerobic activity during the offseason phase to maintain aerobic conditioning and vagal control”. How can you affirm what is said in these lines? These conclusions are not derived from the results of this work.

- Please, better describe and highlight the practical implications of this work

Author Response

Thank you for your comments. We have made significant changes to the manuscript and provided the responses in the attached document.

Reviewer 2 Report

Comments and Suggestions for Authors

Comments to Authors:

The article has a low scientific level in the elaboration of the text. My comments are:

·       The structure of the abstract is not correct. Authors must follow the journal's guidelines: objective, methods,... 

·       Line 20-23: Authors must rewrite the results of the study in the abstract. It must be better explained.

·       Line 67-70: The goal must be clearer, better writing

·       Why have the authors not analyzed the Poincare graphic and its SD1, SD2, and ratio? The studied parameters provides them on a device, or the authors analyzed the data series.

·       I think the authors must include the analysis of the Poincare graphic

·       Point 3.1 must go in the method section, at point 2.1

·       The value P of the age is not in the table

·       In the discussion section, they should not repeat the goal. The authors must begin by providing the study finding.

·       Authors must include a "Clinical Application" section within the discussion section.

·       The authors must change the conclusion. It should be concise, clear and a statement that reflects the most important thing in this paper. The conclusion should be 3-4 lines.

·       Reference style is not correct: 1,3,4,… they review all them.

Comments on the Quality of English Language

Moderate editing of English language required

Author Response

Thank you for your comments. We made significant changes to the manuscript and have provided our responses below.

Round 2

Reviewer 1 Report

Comments and Suggestions for Authors

Dear authors, thank you for your comprehensive response to all my comments. Now the article has improved especially in the description of the methods and practical applications of the study. Given the sample size, please consider whether to add “a pilot study” in the title.

Reviewer 2 Report

Comments and Suggestions for Authors

The authors have responded to my comments satisfactorily.